# LEARNING IN REVERSE CAUSAL STRATEGIC ENVIRONMENTS WITH RAMIFICATIONS ON TWO SIDED MARKETS

**Seamus Somerstep**
Department of Statistics
University of Michigan
smrstep@umich.edu

**Yuekai Sun**
Department of Statistics
University of Michigan
yuekai@umich.edu

**Ya'acov Ritov**
Department of Statistics
University of Michigan
yritov@umich.edu

## ABSTRACT

Motivated by equilibrium models of labor markets, we develop a formulation of causal strategic classification in which strategic agents can directly manipulate their outcomes. As an application, we compare employers that anticipate the strategic response of a labor force with employers that do not. We show through a combination of theory and experiment that employers with performatively optimal hiring policies improve employer reward, labor force skill level, and in some cases labor force equity. On the other hand, we demonstrate that performative employers harm labor force utility and fail to prevent discrimination in other cases.

## 1 INTRODUCTION

In many applications of predictive modeling, the model itself may affect the distribution of samples on which it has to make predictions; this problem is known as strategic classification (Hardt et al., 2015; Brückner et al., 2012) or performative prediction (Perdomo et al., 2020). For example, traffic predictions affect route decisions, which ultimately impact traffic. Such situations can arise in a variety of applications; a common theme is that the samples correspond to strategic agents with an incentive to "game the system" and elicit a desired outcome from the model.

In the standard strategic classification setup, the agents are allowed to modify their features, but they do not modify the outcome that the predictive model targets. An example of this is spam classification: spammers craft their messages (*e.g.* avoiding certain tokens) to sneak them past spam filters. There is a line of work on *causal strategic classification* that seeks to generalize this setup by allowing the agents to change both their features and outcomes, usually by incorporating a causal model between the two (Miller et al., 2020; Kleinberg and Raghavan, 2020; Haghtalab et al., 2023; Horowitz and Rosenfeld, 2023).

In this paper, motivated by equilibrium models of labor markets (see Fang and Moro (2011a) for a survey), we study a strategic classification setup in which the agents are able to manipulate their attributes via a *reverse causal* mechanism. This complements prior work on causal strategic classification, in which the agents manipulate their attributes via causal mechanisms. As an example of reverse causal strategic classification, we consider the employer's problem in labor market models. In particular, we study the consequences (in terms of employer and labor force welfare) of hiring policies that anticipate reverse causal strategic responses (we will refer to such anticipatory policies as strategic and later performative or optimal).

1. In the simple Coate and Loury (1993) labor market model, we show theoretically that such strategic policies lead to higher employer rewards (compared to non-strategic hiring policies). Thus, rational employers should be performative. Further, such hiring policies improve labor force skill level and equity, so performative employers also benefit the labor force.

2. Unfortunately, we also observe that in some aspects, the desirable properties of (reverse causal) strategic hiring policies are brittle. To study their robustness, we developed a more sophisticated general equilibrium labor market model. We show empirically that while our theory generalizes, strategic hiring policies will harm workers by reducing their aggregate welfare and can still lead to disparities amongst labor force participants.

## 1.1 RELATED WORK

The study of learning on data distributions that are dependent on the learned model (known as performative prediction) was formalized in (Perdomo et al., 2020). The works Mendler-Dünner et al. (2020); Drusvyatskiy and Xiao (2020); Wood et al. (2021) extend the problem to a stochastic setting. Perhaps the most prevalent example of performative prediction is strategic classification (Hardt et al., 2015). A work in the strategic classification literature close in spirit to ours is Liu et al. (2022), in which the authors model agents as competing in contests. Our work is also close to the works that inject an element of causality into strategic classification Alon et al. (2020); Kleinberg and Raghavan (2020); Haghtalab et al. (2023); Miller et al. (2020); Shavit et al. (2020); Harris et al. (2022); Horowitz and Rosenfeld (2023); Mendler-Dünner et al. (2022). Crucially, each of these works assumes that strategic agents manipulate features $x$, which in turn has a causal effect on outcomes $y$; we are interested in the reverse: when strategic agents manipulate $y$, which in turn has a causal effect on features $x$.

Our work is inspired by the economic models of statistical discrimination in labor markets (Arrow, 1971; Phelps, 1972; Coate and Loury, 1993; Moro and Norman, 2003; 2004). See Fang and Moro (2011a) for a survey of this work. The works Liu et al. (2020), Kannan et al. (2019) give a contemporary study of Coate and Loury (1993) type models in the context of algorithmic decision-making and affirmative action, respectively.

We provide an extended discussion of related works in the appendix.

## 1.2 PRELIMINARIES

Performative prediction, introduced in Perdomo et al. (2020), seeks to study distribution shifts that are dependent on model deployment. To be more specific, if the learner picks model parameter $\theta$, then the next samples are drawn from $\mathcal{D}(\theta)$. Thus, a performative learner seeks to minimize

$$\mathrm{PR}(\theta) \triangleq \mathbf{E}_{Z \sim \mathcal{D}(\theta)}[\ell(Z; \theta)].$$

This problem can be non-convex; most works assume the learner utilizes a strategy of repeatedly retraining a model. Under this strategy, at round $t + 1$ of training, the learner deploys

$$\theta_{t+1} = \arg \min_{\theta'} \mathbf{E}_{Z \sim \mathcal{D}(\theta_t)}[\ell(Z; \theta')].$$

The authors of Perdomo et al. (2020) show that this strategy (known as repeated risk minimization) will converge to stable points, which are defined by:

$$\theta_{\mathrm{stab}} \in \arg \min_{\theta'} \mathbf{E}_{Z \sim \mathcal{D}(\theta_{\mathrm{stab}})}[\ell(Z; \theta')].$$

The second performative solution concept is a performatively optimal point, which satisfies

$$\theta_{\mathrm{opt}} \in \arg \min_{\theta'} \mathbf{E}_{Z \sim \mathcal{D}(\theta')}[\ell(Z; \theta')].$$

It is worth emphasizing that optimal points are generally not stable, and stable points are generally not optimal; in particular, stable points are not minima of the performative loss, and optimal points need not be the best response to their own induced distribution.

The canonical example of performative prediction is strategic classification.

**Example 1.1** (Hardt et al. (2015)). *Although it predates performative prediction, strategic classification Hardt et al. (2015) can be viewed as an instance of performative prediction in which users game their features. More concretely, there are users with features $x \in \mathcal{X}$, discrete outcomes $y \in \mathcal{Y} \simeq [K]$, and a corresponding base distribution $P$ over $\mathcal{Z} = \mathcal{X} \times \mathcal{Y}$. The goal of learning is to deploy a model $f : \mathcal{X} \to \mathcal{Y}$ using training data drawn from $P$. The catch is the following: post-training and pre-testing of $f$ users will game their features (at some cost $c : \mathcal{X} \times \mathcal{X} \to \mathbb{R}^+$) via the update rule:*

$$x \to x_f \triangleq \arg \max_{x' \in \mathcal{X}} f_\theta(x') - c(x, x').$$

*The goal is to deploy a classifier that is accurate post-distribution shift. The ideal classifier minimizes $\mathbb{E}_P \ell(f(x_f), y)$.*

## 2 REVERSE CAUSAL STRATEGIC LEARNING

In the standard strategic classification setting, agents update their features $x$, but they do not update their labels $y$. This is common when the agents wish to "game" the model; the standard example is spam classification: spam creators change their emails to try and avoid spam filters (for example, by avoiding certain tokens), but their emails will remain spam.

Recently, Horowitz and Rosenfeld (2023) considered a more general problem setting by permitting agents to modify their labels (in addition to their features) through a causal/structural model. Here, the agents still change their features, but changes to the features can propagate through the causal model and (indirectly) lead to changes in their labels. In the following, we consider a *reverse causal* setting in which the agents change their labels and the changes propagate through a structural model to their features. Here are two motivating examples for the reverse causal strategic learning setting: one from labor economics and one from game models of content platforms.

**Example 2.1** (Coate and Loury (1993)). *Consider an employer that wishes to hire skilled workers. The worker skill level is represented as $Y \in \{0, 1\}$ with $Y \sim Ber(\pi)$. The employer implements a (noisy) skill level assessment, with the outcome represented as $X \in [0, 1]$. The employer receives utility $p_+$ from a "qualified" hire and suffers loss $p_-$ from an "unqualified" hire, and seeks to train a classifier $f(x) : [0, 1] \to \{0, 1\}$ that optimizes their overall utility:*

$$\max_{f \in \mathcal{F}} \left[ p_+ \pi \mathbb{P}(f(x) = 1 \mid y = 1) - p_-(1 - \pi) \mathbb{P}(f(x) = 1 \mid y = 0) \right],$$

*Thus far, this is a standard classification problem because the workers are non-strategic, i.e. they are unaffected by the employer's hiring policy. To introduce a strategic component, the workers are allowed to become qualified (at a cost) in response to the employer's policy. Let $w > 0$ be the wage paid to hired workers and $c$ be the (random and drawn from CDF $G$) cost to a worker of becoming skilled. For an unskilled worker, the expected utility of becoming skilled is*

$$u_w(f, y) \triangleq \begin{cases} \int_{[0,1]} w f(x) d\Phi(x \mid 1) - c & \text{if the worker becomes skilled,} \\ \int_{[0,1]} w f(x) d\Phi(x \mid 0) & \text{if the worker remains unskilled,} \end{cases}$$

*where $\Phi(x \mid y)$ is the conditional distribution of skill level assessments. So a strategic worker becomes skilled if*

$$w \int_{[0,1]} 1_{\{f(x)=1\}} d\Phi(x|Y = 1) - w \int_{[0,1]} 1_{\{f(x)=1\}} d\Phi(x|Y = 0) - c > 0.$$

**Example 2.2** (Hron et al. (2023), Jagadeesan et al. (2023)). *Consider a social media system. There is a demand distribution $u \sim P_d; u \in \mathbb{R}^d$ from which users arrive to the system sequentially, with $u^t$ representing a vector of characteristics of the $t'th$ user. There is also a population of $n$ content producers which each produce content $\{s_i^t\}_{i=1}^n; s_i^t \in \mathbb{R}^d$. Associated with a piece of content $s$ is some noisy measurement $X \in \mathbb{R}^d$ of the content generated via CDF $\Phi(\cdot \mid s)$. For example, $X$ could consist of the sentiment of "comment" $s$ receives from users, as well as the amount of likes and views $s$ generates. In general, we interpret $\mathbb{E}||X||_2^2$ as the "amount" of the content that a creator with signal $X$ produces. The learner wishes to train a recommendation system $R(\Sigma, x, u)$ that recommends content $s_i^t$ to the user $u_t$ with probability:*

$$P(u_t \text{ is assigned content } s_i^t) = R(\Sigma, x_i^t, u^t) = \frac{\exp(\frac{1}{\tau}(x_i^t)^T \Sigma u^t)}{\sum_{j=1}^n \exp(\frac{1}{\tau}(x_j^t)^T \Sigma u^t)}$$

*The learner wishes to maximize the true enjoyment a user receives from content, which is given by*

$$r(\Sigma^*, u^t, s^t) = (s^t)^T \Sigma^* u^t.$$

*Content producers are not static, however, and will adapt content production. Since exposure to users is directly connected to the learned recommendation system, a content creator with original content $s$ will tend towards producing content $s'$ that optimizes*

$$\mathbb{E}_{u \sim P_d; x \sim \Phi(\cdot|s')} R(\Sigma, x, u) - \mathbb{E}_{x \sim \Phi(\cdot|s')}||x||_2^2$$

### 2.1 THE REVERSE CAUSAL STRATEGIC LEARNING PROBLEM

In reverse causal strategic learning, the samples $(X, Y)$ are agents, and the learner wishes to learn a (possibly randomized) policy/rule $f : \mathcal{X} \to \mathcal{Y}$ to predict the agents' responses $Y$ from their features

$X$. The agents are fully aware of the learners policy $f$ and as such, we assume that the response of the agents to $f$ is to change their outcomes $Y$ strategically:

$$Y \rightarrow Y_+(f, Y) \triangleq \arg\max_{y'} W(f, y') - c(y', Y), \qquad (2.1)$$

where $W$ is a welfare function that measures the agent's welfare and $c$ is a (possibly random) cost function that encodes the cost (to the agent) of changing their outcome from $Y$ to $y'$.

**Example 2.3** (Example 2.1 cont). *In the labor market example, the agents are the workers, and their welfare is their expected wage*

$$W(f, y') = \int_{[0,1]} w f(x) d\Phi(x \mid y'); y' \in \{0, 1\},$$

*while the cost is*

$$c(y', y) = cy'; \ c \sim G, \ y' \in \{0, 1\}.$$

**Example 2.4** (Example 2.2 cont). *In the content creation example, the agents are the content producers. Their welfare function is*

$$W(\Sigma, s') = \mathbb{E}_{u \sim P_d; x \sim \Phi(\cdot \mid s')} R(\Sigma, x, u),$$

*while the cost function is simply the effort $\mathbb{E}_{x \sim \Phi(\cdot \mid s')} \|x\|_2^2$ required to produce a certain "amount" of content.*

The key ingredient that makes this setting *reverse causal* is that strategic change in the outcome propagates to cause a change in the generation of features $X$ via:

$$X \sim \Phi(\cdot \mid Y) \rightarrow X_+(f, Y) \sim \Phi(\cdot \mid Y_+(f, Y)),$$

where $\Phi(\cdot \mid Y = y)$ is the conditional distribution of the features $X$ given the outcome $Y$ in the base distribution of the agents. In other words, the agents are unable to directly change their features in the reverse causal setting; *they can only change their features by changing their outcomes.* This distinguishes the reverse causal strategic learning setting from the problem settings in other works on performative prediction and strategic classification (Hardt et al., 2015; Horowitz and Rosenfeld, 2023).

Going back to the learner, their welfare depends on the post-response agents, so they must account for the strategic response of the agents. A learner which does this is *performative* and minimizes the post-strategic response loss:

$$\min_{f \in \mathcal{F}} \mathbf{E}\big[\ell(f(X_+(f, Y)), Y_+(f, Y))\big]. \qquad (2.2)$$

In the rest of the paper, we will return to the labor market, demonstrating that a learner with a proactive strategy is often mutually beneficial for both the learner and strategic agents. A discussion on the minimization of Objective 2.2 is reserved for appendix A in which we provide the a simple algorithm for the learner to minimize the performative loss in a stochastic setting.

## 3 STRATEGIC HIRING IN THE COATE-LOURY MODEL

In this section, we focus on example 2.1 and study the impact of performatively optimal hiring policies on employer and labor force welfare in the Coate-Loury model. Despite the strategic nature of the labor force, prior studies of labor market dynamics generally assume employers are merely reactive (instead of proactive) to the strategic responses of the labor force (Fang and Moro, 2011a). We show that in a variety of market settings, performatively optimal hiring policies improve employer welfare, labor force welfare, and labor market equity.

### 3.1 PERFORMATIVE PREDICTION IN THE COATE-LOURY MODEL

Recall the setup from example 2.1. We impose some standard assumptions on the problem:

1. $\frac{P(X=x|y=1)}{P(X=x|y=0)} = \frac{\phi(x|y=1)}{\phi(x|y=0)}$ is monotonically increasing in $x$,

2. $\phi(x \mid y)$ is continuously differentiable for $y \in \{0, 1\}$,

3. $G(0) = 0$, $G$ is continuously differentiable, and $c \sim G$ is almost surely bounded above by $M_G$.

The second two assumptions are technical, but the first is more substantive. It resembles the monotone likelihood ratio assumption in statistical hypothesis testing that ensures consistency of hypothesis tests. In this case, it ensures the optimal hiring policy is a threshold policy of the form $\mathbf{1}\{x \geq \theta\}$ for some $\theta$, this is the operating assumption in the original Coate and Loury (1993) paper, and we will proceed with it as well.

Armed with the assumption of a threshold hiring policy, we discuss performative prediction in the Coate and Loury (1993) model. Recall the strategic response of the labor force participants in example 2.1. In aggregate, the proportion of skilled labor force participants becomes

$$\pi(\theta) = G(w[P(x > \theta \mid y = 1) - P(x > \theta \mid y = 0)]).$$

Given this, the employer's strategic hiring problem is really an instance of performative prediction with the performative employer utility given by

$$U_{\text{perf}}(\theta) \triangleq p_+ \mathbb{P}(X > \theta \mid y = 1)\pi(\theta) - p_- \mathbb{P}(X > \theta \mid y = 0)(1 - \pi(\theta)).$$

As in other performative settings, the employer may opt to deploy an optimal or stable policy. An employer that deploys an optimal policy anticipates and accounts for the labor force participants responses to their decisions and thus deploys a policy that solves the performative problem

$$\theta_{\text{opt}} \in \arg\max_\theta U_{\text{perf}}(\theta). \tag{3.1}$$

An employer that deploys a stable policy is reactive rather than anticipatory towards labor force strategic responses, *i.e.* they deploy a stable policy, which is any policy that maximizes employer utility on its own induced distribution:

$$\theta_{\text{stab}} \in \arg\max_\theta p_+ \pi(\theta_{\text{stab}})\mathbb{P}(X > \theta \mid y = 1) - p_-(1 - \pi(\theta_{\text{stab}}))\mathbb{P}(X > \theta \mid y = 0).$$

The concept of performative vs stable solutions in the context of a micro-economic model has a game theoretic interpretation. A performative solution is the Stackleberg equilibrium to a 2-player game between the firm and the labor force, with the firm as the leader and the labor force as the follower. On the other hand, the stable solution is the Nash equilibrium between the firm and the labor force if there is no leader-follower dynamic.

We will study both low-wage and high-wage markets. Low-wage markets possess the desirable property that any employer that follows a "greedy" strategy, *i.e.* they sequentially deploy

$$\theta_{t+1} \leftarrow \arg\max_\theta p_+ \mathbb{P}(X > \theta \mid y = 1)\pi(\theta_t) - p_- \mathbb{P}(X > \theta \mid y = 0)(1 - \pi(\theta_t)), \tag{3.2}$$

will eventually converge on a stable policy. This is because (3.2) is an instance of RRM, and low worker wages (in addition to some regularity conditions) will ensure that the requirements for RRM convergence provided in Perdomo et al. (2020) are satisfied (see Appendix D).

High-wage markets are independently interesting, as in such markets the benefits of optimal policies for a firm will be substantial. On the other hand, in high-wage markets, the convergence of RRM may not be universal. This is not necessary for any of our results (stable policies will always *exist* and any greedy employer that does stabilize will do so on a stable policy). Additionally, our empirical results also demonstrate that, in practice, convergence of reactive firms is not sensitive to market conditions.

Finally, we remark that each theorem will need additional assumptions on the structure of the labor market (for example, worker wage and firm reward). The social situations that the Coate and Loury (1993) model applies to are broad Fang and Moro (2011b); and thus the applicability of each theorem will depend on the social environment at hand. For any specific social situation, a full justification of these assumptions would require a study of the model on real labor market data (or the alternative social situation). Such studies are relatively rare in the literature, but two examples are Arcidiacono et al. (2011) (studies admissions into Duke), Altonji and Pierret (2001) (studies education data).

### 3.2 EFFECTS OF STRATEGIC HIRING ON EMPLOYER AND LABOR FORCE WELFARE

The main result of this subsection compares employer welfare (in terms of the employer's expected utility) and labor force welfare (in terms of the fraction of skilled labor force participants) resulting from optimal and stable hiring policies. We impose two additional assumptions on the market.

1. there exists $\tilde{\theta} \in [0, 1]$ such that $P(X > \tilde{\theta} \mid y = 1) - P(X > \tilde{\theta} \mid y = 0) > \delta_1 > 0,$

2. $\frac{\phi(x|1)}{\phi(x|0)} > \delta_2 > 0$ for any $x \in [0, 1]$.

Together, these ensure that some distinction between skilled and unskilled workers is possible, but no policy can perfectly distinguish between the two. These assumptions ensure that stable and optimal policies will be distinct.

We study markets in both the low-wage and high-wage regimes. In the large-wage regime, firms receive substantial benefits from being optimal.

**Theorem 3.1.** *If $w > M_G/\delta_1$ and $p_+ > p_-/(\delta_1\delta_2)$ then the following holds for all stable parameters $\theta_{stab}$ and all optimal parameters $\theta_{opt}$:*

1. $\pi(\theta_{opt}) > \pi(\theta_{stab})$.

2. $U_{perf}(\theta_{stab}) \leq U_{perf}(\theta_{opt})/(1 + \delta_1)$.

On the other hand, in the low-wage regime, the firm's benefits from an optimal strategy may be small.

**Theorem 3.2.** *If $w > 0$, $p_+ > max(1, \frac{\pi(\tilde{\theta})}{(1-\pi(\tilde{\theta}))}p_-)$ and $\delta_1\delta_2 > p_-$, then the following holds for all stable parameters $\theta_{stab}$ and all optimal parameters $\theta_{opt}$:*

1. $\pi(\theta_{opt}) > \pi(\theta_{stab})$.

2. $U_{perf}(\theta_{stab}) \leq U_{perf}(\theta_{opt})$.

To see the intuition behind Theorems 3.1 and 3.2, consider an extreme case where $p_- = 0$. Here, since for any $\pi > 0$, the best reactive employer response is $\theta = 0$, any stable policy will satisfy $\pi(\theta_{stab}) = 0$ (and thus $U_{perf}(\theta_{stab}) = 0$), while clearly an optimal employer can do better than this and will in general deploy a model that results in $\pi(\theta_{opt}) > 0$. Theorems 3.1 and 3.2 are a generalization of these simple dynamics.

The alignment between the social welfare of the workers and the reward to the learner is in contrast with other areas of strategic learning. For example, Milli et al. (2018) show that there is a direct trade-off between the social burden on strategic agents and learner accuracy in strategic classification.

### 3.3 Effects of Strategic Hiring on Labor Force Equity

Besides labor force skill level, another pressing issue in labor markets is equity. In fact, the Coate-Loury model was developed to show how inequities may arise in labor markets despite the lack of explicit discrimination in the market. First, we recall the original two-group version of the Coate-Loury model.

Employers now seek to hire workers from a large population of labor force participants consisting of two identifiable groups denoted as Maj or Min, with $\lambda$ denoting the fraction of workers with a Maj group membership. Crucially, employer profits, worker costs, wages, and worker signals are agnostic with respect to group membership. Denoting the proportion of qualified workers in a group as $\pi^{\text{Maj}}$ and $\pi^{\text{Min}}$ the (non-performative) employer utility is:

$$U(\theta^{\text{Maj}}, \theta^{\text{Min}}) = \lambda U(\theta^{\text{Maj}}) + (1 - \lambda)U(\theta^{\text{Min}}),$$

The population level response of labor force participants to deployed policies is largely similar in the group case:

$$\begin{aligned}
\pi^{\text{Maj}}(\theta^{\text{Maj}}, \theta^{\text{Min}}) &= G(w[P(x > \theta^{\text{Maj}} \mid y = 1) - P(x > \theta^{\text{Maj}} \mid y = 0)]), \\
\pi^{\text{Min}}(\theta^{\text{Maj}}, \theta^{\text{Min}}) &= G(w[P(x > \theta^{\text{Min}} \mid y = 1) - P(x > \theta^{\text{Min}} \mid y = 0)]).
\end{aligned} \quad (3.3)$$

Note that under these assumptions, both the employer's non-performative hiring problem and the performative one are seperable; *i.e.* the employer can simply solve two hiring problems for each group separately. As such, we define a stable pair $\vec{\theta}_{\text{stab}} = (\theta_{\text{stab}}^{\text{Maj}}, \theta_{\text{stab}}^{\text{Min}})$ as a pair of policies that are each stable and optimal pairs $\vec{\theta}_{\text{opt}} = (\theta_{\text{opt}}^{\text{Maj}}, \theta_{\text{opt}}^{\text{Min}})$ as a pair of policies that are each (performatively) optimal.

In Coate and Loury (1993) a policy pair $\vec{\theta}$ is discriminatory if $\pi^{\text{Maj}}(\vec{\theta}) \neq \pi^{\text{Min}}(\vec{\theta})$, this will also be our metric for discrimination in strategic hiring. It is worth emphasizing that our definition of a discriminatory stable pair is exactly the definition of a discriminatory market equilibrium in Coate and Loury (1993).

We will see that, in a certain sense, an employer with a performatively optimal strategy enforces fairness amongst the groups, while a reactive employer provides no such guarantees. Some technical assumptions are needed on the conditions of the market; as the employer's hiring problem is separable with respect to group, we state these assumptions in the context of a single group market, with the implication that the requirements hold when the market is constrained to either group.

We will study markets with the following assumptions:

1. There exists $\tilde{\theta}$ such that $P(x > \tilde{\theta}|y = 1) - P(x > \tilde{\theta}|y = 0) > 1 - \epsilon$

2. $p_+ = p_-$

The second assumption is primarily to simplify the problem. On the other hand, the first assumption is crucial. It implies that the firm find a hiring policy that provides excellent separation between qualified and unqualified workers (generally $\epsilon$ can be thought of as small). Under such an assumption, a performative firm will have both ability and motive to steer the market towards equitable equilibrium.

**Theorem 3.3.** *Assume $\theta^{*-1}(\theta)$ is $c-$Lipschitz on $[0, \tilde{\theta}]$, where $\theta^*(\pi)$ is the best employer response if $P(y = 1) = \pi$ and that $w > M_G/(1 - \epsilon)$ Then the following hold simultaneously:*

1. *There exists a stable pair $\vec{\theta}_{stab}$ such that $|\pi^{Maj}(\vec{\theta}_{stab}) - \pi^{Min}(\vec{\theta}_{stab})| > 1 - c$.*

2. *$|\pi^{Maj}(\vec{\theta}_{opt}) - \pi^{Min}(\vec{\theta}_{opt})| < \epsilon$ for all optimal pairs $\vec{\theta}_{opt}$.*

The first assumption requires sufficient smoothness on the inverse of the firm's best response; in general, it is easy to construct markets where both $c$ and $\epsilon$ are small. Such an example is presented in Appendix B, and some sufficient market conditions for these conditions to hold are provided in Appendix C.

We also study the problem in low-wage markets, a strong concavity assumption on the firms non-performative utility $U(\theta)$ is needed. This is generally not too difficult to meet, and a discussion is supplied in Appendix D.

**Theorem 3.4.** *Assume that the non-performative firm utility $U(\theta)$ is $\gamma-$ strongly concave for all $\pi$. Also, assume $g(\cdot)$, and $\phi(\cdot())$ are bounded above by $K_1$, and they are differentiable and $g'(\cdot)$, $\phi'(\cdot)$ are bounded above by $K_2$. If $G^{-1}(\frac{\phi(\tilde{\theta}|0)}{(1-\epsilon)(\phi(\tilde{\theta}|1)+\phi(\tilde{\theta}|0))}) < w < \gamma/(2K_1K_2)$ then the following hold simultaneously:*

1. *There exists a stable pair $\vec{\theta}_{stab}$ such that $|\pi^{Maj}(\vec{\theta}_{stab}) - \pi^{Min}(\vec{\theta}_{stab})| > 0$.*

2. *$|\pi^{Maj}(\vec{\theta}_{opt}) - \pi^{Min}(\vec{\theta}_{opt})| = 0$ for all optimal pairs $\vec{\theta}_{opt}$.*

For this to truly be a "low-wage market" we need $G^{-1}(\frac{\phi(\tilde{\theta}|0)}{(1-\epsilon)(\phi(\tilde{\theta}|1)+\phi(\tilde{\theta}|0))}) \approx 0$, we will see an example of markets that satisfy this and the other assumptions in the appendix.

The intuition behind optimal policies enforcing fairness in Theorems 3.3 and 3.4 is relatively simple; a performative employer that can distinguish skill level well posseses both direct incentive and ability to ensure that both majority and minority group worker qualification levels will be high, and as such the discriminatory gap will be small. On the other hand, a reactive employer has no direct incentive to increase the fraction of skilled workers; *i.e.* they are too myopic to maximize the fraction of skilled workers. Thus, there is no invisible hand steering the market to more equitable equilibria.

## 4    EXPERIMENTS ON MARKETS WITH CONTINUOUS SKILL LEVELS

In this section, we study the sensitivity of the promising labor force skill and equity results in the preceding section to the underlying Coate-Loury model. A key limitation of the two-group Coate-Loury model is that the labor markets for the two groups operate independently of each other. Here

we study the problem using a more sophisticated general equilibrium model (introduced in Appendix B) which allows for such cross-group effects. In our formulation of a labor market, we empirically observe the following economic takeaways:

1. As in the Coate-Loury model, the employer is always incentivized to be performative. Also, a performative employer benefits labor force participants because it increases labor force skill levels.

2. Unfortunately, a performative employer harms workers by reducing their aggregate welfare. Additionally, the fairness benefits of a performative employer are brittle with respect to the assumptions of the underlying market.

Due to space limitations, we only briefly describe our modifications to the Coate-Loury model here and defer a detailed description of the market to Appendix B. The first is that worker skill level $Y$ has Lebesgue density $p(y)$, implying that qualification is not binary, but rather a continuum of possible productivity levels. The second modification is to the reverse causal updates of the workers; while they remain strategic and update their outcomes according to the reverse causal mechanism 2.1, the wage structure and cost are different. The cost is now a fixed function: $c(\cdot, \cdot) : \mathcal{Y} \times \mathcal{Y} \to \mathbb{R}^+$. The wage and utility structure will be more complex, again we refer to Appendix B for a detailed description of changes.

### 4.1 EXPERIMENTS WITH LINEAR UTILITY AND FLAT WAGES

In the Coate and Loury (1993) model, it is assumed that production functions are linear and that the wage structure is flat. In our context, this is the case where $w(x, f) = w$ and $u(y, H_f(y)) = H_\pi(y)(ay - 1)$. Finally, the cost is specified to be quadratic: $c(y', y) = \frac{c}{2}(y' - y)^2$.

To study worker welfare and employer utility, we assume $p(y)$ is known, so the employer's policy optimization is non-stochastic. Figure 1 diagrams the worker side. In general, an optimal policy results in a larger portion of qualified workers, while a stable policy leads to greater aggregate worker welfare. The right-hand side of Figure 2 plots the firm side; unsurprisingly, the employer prefers an optimal policy; additionally, the left-hand side of Figure 2 demonstrates algorithm 1 (appendix A) in a labor market.

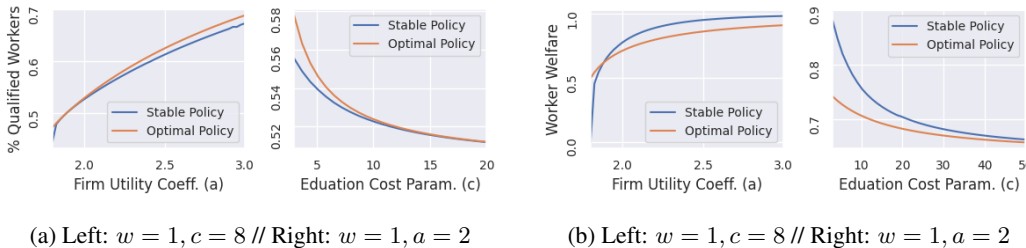

(a) Left: $w = 1, c = 8$ // Right: $w = 1, a = 2$     (b) Left: $w = 1, c = 8$ // Right: $w = 1, a = 2$

Figure 1: Worker Welfare and Proportion of Qualified Workers for Optimal and Stable Policies

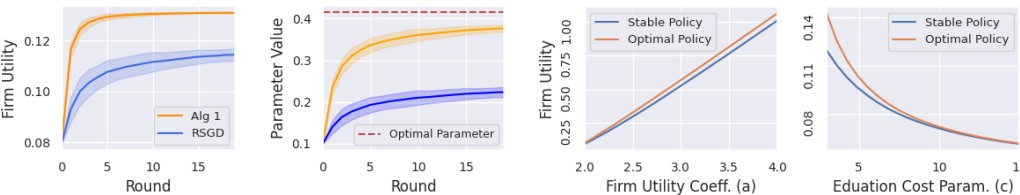

Figure 2: Left: RSGD and Alg 1 ($a = 2, w = 1, c = 5$) // Right: employer Utility Under Policies (Left: $w = 1, c = 5$ // Right: $w = 1, a = 2$)

## 4.2 EXPERIMENTS WITH NON-LINEAR UTILITY AND NON-FLAT WAGES

Next, we present experiments on markets that include identifiable groups and move beyond the assumptions of linear utility and flat wages. In the subsequent experiments, wages are non-linear and are determined by a Nash equilibrium among competing employers (inspired by (Moro and Norman, 2004)). Additionally, we assume $u(y, H_f(y)) = y(H_f(y))^\alpha$, the important difference being that $u(\cdot, \cdot)$ is non-linear in hiring probability, which removes the separability property present in the Coate-Loury model. Finally, we no longer assume that cost is agnostic with respect to group; cost is now specified by $c(y', y, i) = \frac{c_i}{2}(y' - y)^2; i \in \{\mathrm{Maj}, \mathrm{Min}\}$.

Figure 3 diagrams the welfare of each group under different policies and parameters; in the top row, group costs and proportions are adjusted (costs are adjusted so total cost remains constant), while in the bottom row, the utility parameter $\alpha$ is adjusted.

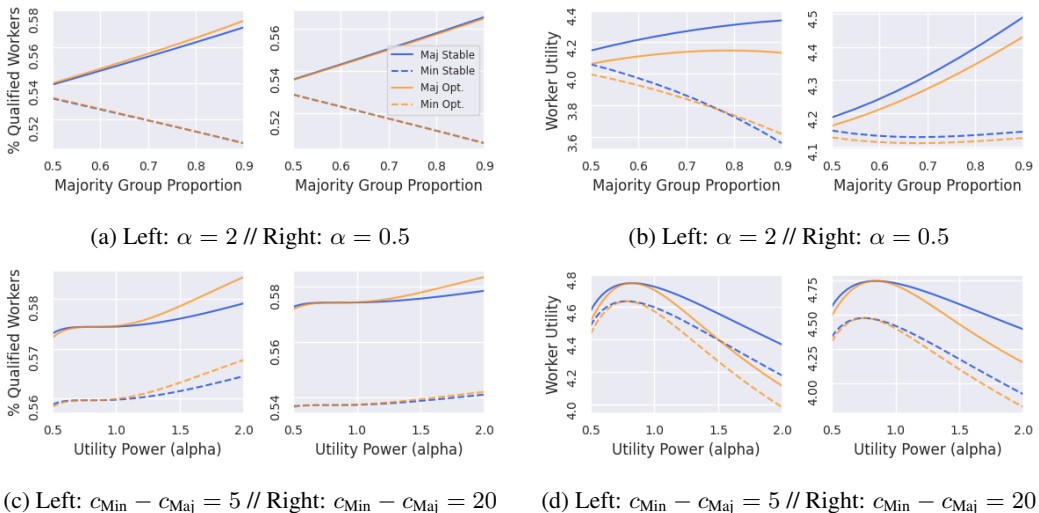

(a) Left: $\alpha = 2$ // Right: $\alpha = 0.5$      (b) Left: $\alpha = 2$ // Right: $\alpha = 0.5$

(c) Left: $c_{\mathrm{Min}} - c_{\mathrm{Maj}} = 5$ // Right: $c_{\mathrm{Min}} - c_{\mathrm{Maj}} = 20$      (d) Left: $c_{\mathrm{Min}} - c_{\mathrm{Maj}} = 5$ // Right: $c_{\mathrm{Min}} - c_{\mathrm{Maj}} = 20$

Figure 3: Worker Welfare and Proportion of Qualified Workers for Optimal and Stable Policies

A similar pattern as before emerges in Figure 3. Workers tend to be more qualified under performative policies (though this is broken in Figure 3 when the utility function is concave in the proportion of qualified workers), but they tend to have a higher average welfare under stable policies. Clearly, optimal policies no longer enforce fairness, as there is no appreciable difference in discrimination between the two.

## 5 CONCLUSION

We have introduced a problem setting that extends strategic classification by allowing agents to directly manipulate $y$, and in turn, cause a shift in $x$; additionally, we provided an algorithm for risk minimization in this setting. As an application of this new framework, we studied the effects of stable and optimal strategies on strategic agent welfare and equity in labor market models. We have demonstrated that a strategic/performative employer can help or harm strategic workers depending on the measurement of agent welfare used. In general, a proactive learner assists the strategic workers in becoming more skilled but harms the strategic agents by reducing their overall aggregate welfare. Additionally, in some but not all cases, a performative learner can assist in preventing discriminatory equilibrium in labor markets. Finally, we have seen that an employer will always benefit from deploying a performative/strategic hiring policy.

### ACKNOWLEDGMENTS

This paper is based upon work supported by the National Science Foundation (NSF) under grants no. 2027737 and 2113373.

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

## A  Stochastic optimization for reverse-causal strategic learning

In this section, we develop a stochastic optimization algorithm to minimize the performative risk (2.2). Previously, the work Izzo et al. (2021) gave an algorithm for performative optimization under the assumption that the performative map is of the form $\mathcal{D}(\theta) = p(z; f(\theta))$, with only $f$ unknown. Our work is in a distinct environment, where the learner has an apriori model for the map $\mathcal{D}$ but the base distribution is unknown. This is similar to the assumptions in Levanon and Rosenfeld (2021), which tackles the optimization for non-causal strategic classification.

### A.1  Learning Set Up

In practice, learning in reverse causal strategic environments is done sequentially (this is similar to the assumed in other performative prediction works). At each round of learning the learner publishes a decision $\theta$ and all data received in that round are drawn via $\mathcal{D}(\theta)$. Succinctly, in a stochastic (or finite data) setting the order of learning at time is as follows:

1. The firm deploys decision $\theta_t$
2. Firm receives data (and corresponding loss/reward) drawn from $\mathcal{D}(\theta_t)$. The firm may use this to either deploy a reactive decision or in some algorithm that converges towards an optimal policy (for example in the methodology illustrated below).

Our methodology will be applicable in this stochastic setting, the two strongest assumptions we need are as follows:

1. As in previous works on optimization for strategic classification (Levanon and Rosenfeld (2021)), the learner is fully cognisant of the reverse causal strategic map $Y_+(y, \theta)$.
2. The learner has knowledge of the feature generating mechanism $\phi(x|y)$. The justification here is that in many reverse causal strategic settings $\phi(x|y)$ is based on some standards that the learner sets (eg a hiring firm giving an interview) and thus the learner should have knowledge of this mechanism

Performative learning in a scenario where $Y_+(y, \theta)$ is unknown is an open and interesting problem. If $\phi(x \mid y)$ is unknown, it can be estimated from data pairs $(x, y)$.

### A.2  Methodology

We let $\phi(x|y)$ denote the conditional density of $x$ given $y$. We also assume that we deploy some parametric model $f_\theta(x)$, and write $Y_+(f_\theta, y) \triangleq G_\theta(y)$. In this notation, the cost function in the anti-causal objective can be rewritten as (recall $\mathcal{Z} = \mathcal{X} \times \mathcal{Y}$)

$$L(\theta) = \int_{\mathcal{Z}} \ell(f_\theta(x), G_\theta(y))\phi(x|G_\theta(y))p(y)dz.$$

We derive the gradient in the following lemma.

**Lemma A.1.** *The gradient of $L(\theta)$ is $\nabla L(\theta) = \nabla_1 L(\theta) + \nabla_2 L(\theta) + \nabla_3 L(\theta)$, where*

$$\begin{aligned}
\nabla_1 L(\theta) &= \int_{\mathcal{Z}} \partial_\theta[f_\theta(x)]\partial_1[\ell(f_\theta(x), G_\theta(y_i))]\phi(x|G_\theta(y_i))dz \\
\nabla_2 L(\theta) &= \int_{\mathcal{Z}} \partial_\theta G_\theta(y_i)\partial_2[\ell(f_\theta(x), G_\theta(y_i))]\phi(x|G_\theta(y_i))dz \\
\nabla_3 L(\theta) &= \int_{\mathcal{Z}} \partial_\theta G_\theta(y_i)\ell(f_\theta(x), G_\theta(y_i))\partial_2[\phi(x|G_\theta(y_i))]dz,
\end{aligned} \quad \text{(A.1)}$$

*and $\partial_\theta G_\theta(y)$ is the solution to the following:*

$$\int_{\mathcal{X}} \partial_\theta[f_\theta(x)]\phi(x|G_\theta(y))dx + \left(\int_{\mathcal{X}} f_\theta(x)\partial_2^2[\phi(x|G_\theta(y))]dx - \partial_1^2 c(G_\theta(y), y)\right)\partial_\theta G_\theta(y) = 0. \quad \text{(A.2)}$$

We propose that the learner utilize Monte-Carlo techniques to numerically approximate the integrals of A.2 and A.1. One applicable option is to use a REINFORCE approximation (see Izzo et al. (2021) or Williams (1992)). This technique allows us to re-write the integrals of A.1, A.2,

$$\int_{\mathcal{X}} f_\theta(x)\partial_2^2[\phi(x|G_\theta(y))]dx = \int_{\mathcal{X}} f_\theta(x)\partial_2^2[\log(\phi(x)|G_\theta(y))]\phi(x|G_\theta(y))dx$$

$$= \mathbb{E}_{x \sim \phi(x|G_\theta(y))}[f_\theta(x)\partial_2^2[\log(\phi(x)|G_\theta(y))]],$$

$$\int_{\mathcal{X}} \ell(f_\theta(x), G_\theta(y_i)) \partial_2[\phi(x|G_\theta(y_i))] dx = \int_{\mathcal{X}} \ell(f_\theta(x), G_\theta(y_i)) \partial_2[\log(\phi(x|G_\theta(y_i)))] \phi(x|G_\theta(y_i)) dx$$
$$= \mathbb{E}_{x \sim \phi(x|G_\theta(y_i))} \ell(f_\theta(x), G_\theta(y_i)) \partial_2[\log(\phi(x|G_\theta(y_i)))].$$

Since the learner knows $\phi$, these expressions can be computed via drawing samples from $\phi(x|G_\theta(y))$. The other integrals in A.2 and A.1 are readily computable via this method in their vanilla forms. We summarize the proposed algorithm for the performative prediction problem in Algorithm 1.

---

**Algorithm 1** Reverse Causal Strategic SGD

---

*Input:* $u, \eta, n, \theta_0$
**while** not converged **do**
    *Draw* $\{y_i\}_{i=1}^n \sim p$ and observe $G_\theta(y_i)$
    Draw $\{x_j^i\}_{j=1}^{n'}$ from $\phi(x|G_\theta(y_i))$
    Compute $\hat{\nabla} L(\theta)(y_i, \{x_j^i\}_{j=1}^{n'})$ with REINFORCE
    $\theta_t \leftarrow \theta_{t-1} - \eta \sum_i \hat{\nabla} L(\theta)(y_i, \{x_j^i\}_{j=1}^{n'})$
**end while**

---

Figure 5 gives an example of this algorithm utilized in the hiring of strategic workers.

### A.3   Proof of Lemma A.1

The only tricky term to evaluate is $\partial_\theta G_\theta(y)$. To deal with problem we utilize the implicit function theorem. We have

$$\partial_\theta G_\theta(y) = \partial_\theta \, argmax_{y'} \int_{\mathcal{X}} f_\theta(x) \phi(x|y') dx - c(y', y).$$

Via the implicit function theorem $\partial_\theta G_\theta(y)$ must solve

$$\partial_\theta[\partial_y(\int_{\mathcal{X}} f_\theta(x)\phi(x|y')dx - c(y', y)) = 0].$$

Which implies that $\partial_\theta G_\theta(y)$ is given by:

$$\int_{\mathcal{X}} \partial_\theta[f_\theta(x)]\phi(x|G_\theta(y))dx + (\int_{\mathcal{X}} f_\theta(x)\partial_2^2[\phi(x|G_\theta(y))]dx - \partial_1^2 c(G_\theta(y), y))\partial_\theta G_\theta(y) = 0. \quad \text{(A.3)}$$

## B   Market Formulation and Experiments

### B.1   Market Details

This market is conceptually similar to that of Coate and Loury (1993), with the twist being that worker skill $y$ is now continuous, with $y \in \mathcal{Y} \subset \mathbb{R}$ generated from $p(y)$. The employer still makes hires based on noisy skill assessment $X$ generated from $\Phi(x|y)$ and group membership $g \sim \text{Ber}[\lambda]$. Employer production is specified by a utility function $u$ which depends on the skill level of a worker and the probability that worker is hired; $u : \mathbb{R} \times [0, 1] \to \mathbb{R}$. Letting $H_f(y) = \lambda H_f^{\text{Maj}}(y) + (1 - \lambda) H_f^{\text{Min}}(y)$ with $H_f^i(y) \triangleq \int_{\mathcal{X}} f_i(x) d\Phi(x|y)$, the employer's (non performative) policy maximization objective is given by

$$\max_{f \in \mathcal{F}} \mathbb{E}[u(y, H_f(y))].$$

A hired worker is given wage $w(x, f)$, dependent on the observed skill level and policy choice and they can improve skills at a cost $c(\cdot, \cdot) \to \mathbb{R}^+$; thus workers strategically choose outcomes by selecting

$$Y_+(y, f) = \arg\max_{y'} \int_{\mathcal{X}} w(x, f_i) f_i(x) d\Phi(x|y') - c(y', y).$$

Using the same notation as in section 2, the employer's post strategic response problem is given by

$$\max_{f \in \mathcal{F}} \mathbb{E}[u(Y_+(y, f), H_f(Y_+(y, f)))].$$

In comparing worker well-being under stable and optimal policies, we first must quantify the well-being of the workers. As before, one metric we use is the proportion of qualified workers, which in this case is the probability a worker drawn from a distribution induced by a policy is skilled enough to provide positive employer utility.

We also introduce a second metric, the cost adjusted worker welfare. Towards this, we define the welfare of a worker with a given natural skill $y$ and group membership $i$ as their net benefit in the market post policy deployment and strategic update. Specifically, this is given by

$$W_i(f, y) = \int_{\mathcal{X}} w(x, f_i) f_i(x) d\Phi(x|Y_+(y, f_i)) - c(Y_+(y, f_i), y).$$

We will define the aggregate worker welfare as the expectation of $W_i(f, y)$ with respect to $p(y)$.

To study the techniques discussed in appendix A, we must focus on the case when the optimal policy $f$ is in a parametric policy class, *i.e.* $\mathcal{F} = f(x, \theta); \theta \in \mathbb{R}$. In the appendix, we provide sufficient conditions for this to hold.

**Proposition B.1.** *Assume the ratio $\phi(x|y_1)/\phi(x|y_0)$ is increasing in $x$ for all $y_1 > y_0$, and the utility $u(y)$ is increasing in $y$. Then the optimal policy choice for the employer is some threshold policy $f(x) = 1_{x > \theta}$.*

*Proof.* We denote the marginal density of the signal $x$ as $\nu(x)$ and subscript any distribution that depends on the policy with $f$; also WLOG assume $\mathcal{Y} = [0, 1]$. We can write the employers (non-performative) utility as the following:

$$U(f) = \int f(x)\nu(x) \int u(y)p(y|x)dy dx$$

Thus the optimal policy is of the form:

$$f(x) = 1\{\int u(y)p(y|x)dy > 0\}$$

Thus the lemma statement is equivalent to $\int u(y)p(y|x)dy$ being monotone in $x$. Using integration by parts we re-write this as the following:

$$\int u(y)p(y|x)dy = u(y)P(Y \leq y|X = x)|_0^1 - \int u'(y)P(Y \leq y|X = x)dy$$

$$= u(1) * 1 - u(0) * 0 - \int u'(y)P_f(Y \leq y|X = x)dy$$

Since $u'(y) > 0$ by assumption; we need to show that $P(Y \leq y|X = x)$ is decreasing in $x$ for all $y$. We have the following:

$$P(Y \leq y|X = x) = \int_0^y P(Y = y|X = x)dy = \frac{\int_0^y \phi(x|y)p(y)dy}{\int_0^1 \phi(x|y)p(y)dy}$$

$$= 1/(1 + \frac{\int_0^y \phi(x|y)p(y)dy}{\int_y^1 \phi(x|y)p(y)dy})$$

By assumption $y_0 < y_1$ the ratio $\phi(x|y_0)/\phi(x|y_1)$ is decreasing in $x$. Thus for all $y$ and any distribution $p$ the ratio $\int_0^y \phi(x|y)p(y)dy/\int_y^1 \phi(x|y)p(y)dy$ is monotonically decreasing in $x$. $\square$

## B.2 NON-LINEAR UTILITY AND WAGES

In several alternative labor market models (Arrow, 1971; Moro and Norman, 2004), wage structures are not flat; instead they are determined by competition between competing employers. Additionally, the authors of Moro and Norman (2004) find that moving beyond linear utility functions introduces inter-group interaction to the labor market, which is a key to revealing sources of discrimination.

Throughout this section we will assume that $u(y, H_\pi(y)) = \gamma(H_\pi(y))u(y)$ for strictly increasing functions $\gamma : [0, 1] \to [0, 1]; u : \mathbb{R} \to \mathbb{R}$; and additionally that $\Pi(x; \theta) = 1_{x > \theta}$.

**Limitations of Linear Utility:**

We briefly discuss the limitations of the model if the production function is linear, *i.e.* $\gamma(\cdot)$ is the identity function. In such a case, via the linearity of expectation, one can write the vanilla employer utility objective as

$$\max_\pi \lambda \mathbb{E} H^0_\pi(y)u(y) + (1-\lambda)\mathbb{E} H^1_\pi(y)u(y).$$

Thus, in this case, the employer can simply pick an optimal policy $\pi_i$ for each group separately. Performative stable policies with which $\pi_1(x) \neq \pi_0(x)$ may exist; but one group does not benefit from discrimination of another group. This lack of interaction is why we go beyond the linear utility case.

**Determination of Wage Structure:**

We adhere to the principle (which is shown rigorously in Moro and Norman (2004)) that at a Nash equilibrium among two or more competing employers, the net profits of each employer should be zero. Equivalently: $\mathbb{E}[\gamma(H_f(y))u(y)] - \mathbb{E}[w(x; f)] = 0$. Thus, the wage offered to a worker with signal $x$ and group membership $i$ should be the expected utility of such a worker conditioned on these traits. We can write this for an individual in group $i$ as

$$\mathbb{E}[\gamma(h_{f_i}(y))u(y)|X = x] = \mathbb{E}[\gamma(f_i(x))u(y)|X = x] = \gamma(f_i(x))\frac{\int_\mathbb{R} u(y)\phi(x|y)dp(y)}{\int_\mathbb{R} \phi(x|y)dp(y)}.$$

As such, for this section we will assume the wages proffered by the employer are

$$w_i(f, x) = \gamma(f_i(x))\frac{\int_\mathbb{R} u(y)\phi(x|y)dp(y)}{\int_\mathbb{R} \phi(x|y)dp(y)}. \tag{B.1}$$

**Example B.2.** *As an example of this assumed wage structure, consider a market specified by* $u(y) = y$, $p(y) = \mathcal{N}(0, \sigma_y^2)$, $f(x|y) = \mathcal{N}(y, \sigma_x^2)$. *In this market, a employer (with hiring threshold* $\theta$*) offers wage*

$$w(x, \theta) = \frac{x}{(1+\sigma_x^2/\sigma_y^2)}\mathbf{1}[x \geq \theta].$$

*Wages scale linearly with perceived skill of a worker; additionally, they increase as the profitability-noise ratio increases (a decrease in* $\sigma_x^2/\sigma_y^2$*).*

*Proof.* The wage structure is given by:

$$1_{x\geq\theta}\frac{\int_\mathbf{R} ye^{-(x-y)^2/2\sigma_x^2}e^{-y^2/2\sigma_y^2}dy}{\int_\mathbf{R} e^{-(x-y)^2/2\sigma_x^2}e^{-y^2/2\sigma_y^2}dy} = 1_{x\geq\theta}\frac{\int_\mathbf{R} ye^{(-1/2\sigma_x^2-1/2\sigma_y^2)(y-x/\sigma_x^2(1/\sigma_x^2+1/\sigma_y^2))^2}dy}{\int_\mathbf{R} e^{(-1/2\sigma_x^2-1/2\sigma_y^2)(y-x/\sigma_x^2(1/\sigma_x^2+1/\sigma_y^2))^2}dy}$$

We can multiply the top and bottom row by the needed normalization constants, then the top row is the expectation of a normal random variable with mean $x/(1 + \sigma_x^2/\sigma_y^2))$ and the bottom row integrates to one. $\square$

For Figure 3 we stick with this example of a market; additionally assuming that the cost is again quadratic $c(y', y) = \frac{c}{2}(y' - y)^2$, and that $\gamma(H_f(y)) = (H_f(y))^\alpha$.

## B.3 ADDITIONAL EXPERIMENTS

In figure 4 we present an example of a market that demonstrates the phenomena discussed in Theorems 3.3 and 3.4. The left hand side plots two curves: the dotted one represents proportion of qualified workers resulting from a hiring policy, while the solid one is employers best response as a function of the proportion of qualified workers; intersections of these correspond to stable policies. There are two stable policies with an appreciable gap in worker qualification; simultaneously the performative utility has a unique maximum and thus all pairs of optimal policies are fair.

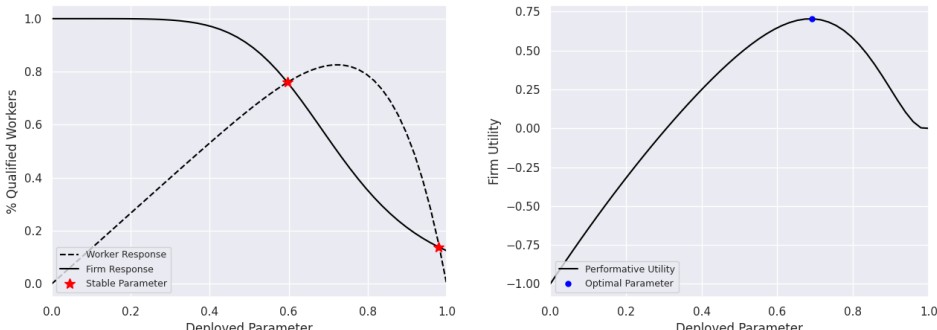

Figure 4: A market with fair optimal pairs and discriminatory stable pairs

Here is another demonstration of Algorithm 1 in the context of labor markets.

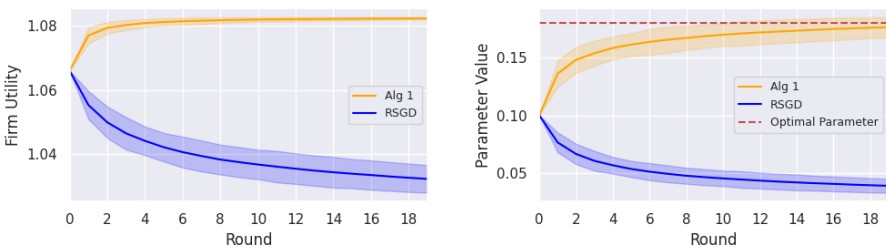

Figure 5: Algorithm 1 and RSGD with $a = 4$, $w = 1$, $c = 4$

### B.4 EXPERIMENTAL DETAILS

**Section 4.2**

1. Figure 2 (left): Experiment is run on a market parameterized by $y_i \sim Unif[0,1]$, $\phi(x|y) = (y+1)x^y 1_{0<x<1}$, $u(y) = ay - 1$, $c(y,y') = \frac{c}{2}(y'-y)^2$ ($a = 2$, $w = 1$, $c = 5$). A total population of 1000 $y$ were drawn and at each round 100 are sampled, learning was run for 20 rounds. 10 separate trials were run, with darker lines showing the mean and the shade indicating a 90 percent confidence interval. Step size $(0.1/(t+1))$ is used for both RSGD and Algorithm 1 iterations, with an initial seed of $0.1$ for each.

2. Figure 1 and figure 2 (right): The same market configuration is used (specific parameters are indicated in captions). Optimal policies were located using sci-py (Virtanen et al. (2020)) minimization packages. To locate stable points RRM was run until iterations were within a distance of $0.001$ of one another.

3. Figure 4: For simplicity, a simpler market parametrization is used with $u(y) = 1_{\{y \geq 0.5\}} - 1_{\{y \leq 0.5\}}$, $\phi(x|y) = 2x 1_{\{y>0.5\}} + 1 * 1_{\{y<0.5\}}$, $p(y) \sim U[0,0.5]$, $c(y,y') = c(y-y')_+$

**Appendix B**

1. Figure 4: $u(y) = 15y$, $p(y) = \mathcal{N}(0,1)$, $\phi(x|y) = \mathcal{N}(y,1)$, $c(y',y) = \frac{c}{2}(y'-y)^2$. For the top row group costs are set as $c_{\text{Min}} = 25/(1-\lambda)$; $c_{\text{Maj}} = 20/(\lambda)$. In the bottom row group costs are set at $c_{\text{Min}} = 25, c_{\text{Maj}} = 20$ or $c_{\text{Min}} = 40, c_{\text{Maj}} = 20$, while $\lambda = 0.8$. Optimal and stable policies are located in a similar manner, with RRM being terminated after iterations are within $0.00001$ of one another for stable polices.

2. Figure 5: Experiment is run on a market parameterized by $y_i \sim Unif[0,1]$, $\phi(x|y) = (y+1)x^y 1_{0<x<1}$, $u(y) = ay - 1$, $c(y,y') = \frac{c}{2}(y'-y)^2$ ($a = 4$, $w = 1$, $c = 4$). A total population of 1000 $y$ were drawn and at each round 100 are sampled, learning was run for 20 rounds. 10

separate trials were run, with darker lines showing the mean and the shade indicating a 90 percent confidence interval. Step size $(0.1/(t+1))$ is used for both RSGD and Algorithm 1 iterations, with an initial seed of 0.1 for each.

## C SECTION 3 PROOFS

### C.1 PROOF OF THEOREM 3.1

*Proof.* For notational convenience we refer to the performative employer utility as $U$. We also denote $\text{TPR}(\theta) = P(X > \theta | y = 1)$ and $\text{FPR}(\theta) = P(X > \theta | y = 0)$ as the true postive rate (resp. false positive rate) of the deployed classifier. The (decoupled) performative learners utility is the following:

$$U(\theta_1, \theta_2) \triangleq p_+ G(w[\text{TPR}(\theta_2) - \text{FPR}(\theta_2)])\text{TPR}(\theta_1) - p_-(1 - G(w[\text{TPR}(\theta_2) - \text{FPR}(\theta_2)]))\text{FPR}(\theta_1)$$

Consider differentiating this wrt the first argument:

$$\partial_1 U(\theta_1, \theta_2) = \partial_{\theta_1}[p_+ \pi(\theta_2) \int_{\theta_1}^1 d\Phi(x \mid 1) - p_-(1 - \pi(\theta_2)) \int_{\theta_1}^1 d\Phi(x|0)]$$

$$\stackrel{FTC}{=} -p_+ \pi(\theta_2)\phi(\theta_1 \mid 1) + p_-(1 - \pi(\theta_2))\phi(\theta_1 \mid 0)$$

By the definition of stability (stable points are optimal on their induced distribution, and end points will not be optimal) any stable point $\theta_s$ must satisfy $\partial_1 U(\theta_1, \theta_2) \mid_{\theta_1 = \theta_2 = \theta_s} = 0$. Thus we must have:

$$-p_+ \pi(\theta_s)\phi(\theta_s|y = 1) + p_-(1 - \pi(\theta_s))\phi(\theta_s|y = 0) = 0$$

Or equivalently:

$$\frac{p_-(1 - \pi(\theta_s))}{p_+ \pi(\theta_s)} = \frac{\phi(\theta_s|1)}{\phi(\theta_s|0)}$$

From here we apply the assumption that $\phi(x|1)/\phi(x|0) > \delta_2$:

$$\frac{p_-(1 - \pi(\theta_s))}{p_+ \pi(\theta_s)} > \delta_2 \implies p_-(1 - \pi(\theta_s)) > \delta_2 p_+ \pi(\theta_s)$$

$$\implies p_- \delta_2 p_+ \pi(\theta_s) + p_- \pi(\theta_s) \implies \frac{p_-}{p_+ \delta_2 + p_-} > \pi(\theta_s).$$

So any stable point $\theta_s$ satisfies $\pi(\theta_s) < \frac{p_-}{p_+ \delta_2 + p_-}$. The intuition from here is straightforward, if $p_+$ is large then $\pi(\theta_s)$ must be small but the optimal labor force skill level for a learner should not be small. To formalize this, consider plugging in the upper bound for $\pi(\theta_s)$ to the learners objective. Any $\theta$ which satisfies the bound will have:

$$U(\theta) \le \frac{p_+ p_-}{p_+ \delta_2 + p_-}\text{TPR}(\theta) \le \frac{p_-}{\delta_2}$$

On the other hand consider $U(\tilde{\theta})$ (with $\tilde{\theta}$ defined as in assumption 1.) Assume that $w$ is large enough so that $w\delta_1 > M_g$, then we have the following:

$$U(\tilde{\theta}) = p_+ * 1 * \text{TPR}(\tilde{\theta}) - p_- * 0 * \text{FPR}(\tilde{\theta}) > p_+ * (\delta_1 + \text{FPR}(\tilde{\theta}))$$

Now clearly if $p_+ > p_-/(\delta_1 \delta_2)$ it holds that $p_+(\delta_1 + \text{FPR}(\tilde{\theta})) > p_-/\delta_2$. In this case, since $U(\tilde{\theta}) > U(\theta)$ for all $\theta$ such that $\pi(\theta) < p_-/(p_+ \delta_2 + p_-)$ it can not hold that $\pi(\theta_{\text{opt}}) < p_-/(p_+ \delta_2 + p_-)$; this completes the proof of the first part. To prove the second part note that $U(\theta_{\text{opt}}) \ge U(\tilde{\theta}) \ge p_+ \delta_1$. Thus:

$$U(\theta_{\text{stab}}) \le \frac{p_- U(\theta_{\text{opt}})}{\delta_1(p_+ \delta_2 + p_-)}$$

Since by assumption $p_+ \delta_1 \delta_2 > p_-$ we have $U(\theta_{\text{stab}}) < p_- U(\theta_{\text{opt}})/(p_- + p_- \delta_1)$ which completes the proof. $\qquad \square$

## C.2   PROOF OF THEOREM 3.2

*Proof.* An identical argument as before will show that any stable point $\theta_s$ satisfies $\pi(\theta_s) < \frac{p_-}{p_+ \delta_2 + p_-}$ which in turn implies that for all stable points $U(\theta_s) < p_-/\delta_2$. Now consider $U(\tilde{\theta})$. We have the following:

$$U(\tilde{\theta}) = p_+ \pi(\tilde{\theta}) \text{TPR}(\tilde{\theta}) - p_-(1 - \pi(\tilde{\theta})) \text{FPR}(\tilde{\theta})$$

By the assumptions of the theorem, we have that $p_+ \pi(\tilde{\theta}) > p_-(1 - \pi(\tilde{\theta}))$, and thus $U(\tilde{\theta}) > \delta_1$. Since $\delta_1 > p_-/\delta_2$, for all $\theta_s$, $U(\tilde{\theta}) > U(\theta_s)$, and in particular no optimal point $\theta_{\text{opt}}$ can have $\pi(\theta_{\text{opt}}) < \frac{p_-}{p_+ \delta_2 + p_-}$.

The statement that $U_{\text{perf}}(\theta_{\text{stab}}) \leq U_{\text{perf}}(\theta_{\text{opt}})$ follows directly from the definition of optimal points.   $\square$

## C.3   PROOF OF THEOREM 3.3

*Proof.* We first prove the statement on stable points. Let $\theta^*(\pi)$ denote the optimal policy for the vanilla utility if the proportion of qualified workers is $\pi$. Note that $\theta^*(\pi)$ is a monotonically decreasing function of $\pi$. The existence of the stable set $\vec{\theta}_{\text{stab}}(\theta_{\text{stab}}^{\text{Maj}}, \theta_{\text{stab}}^{\text{Min}})$ such that $\pi^{\text{Maj}}(\vec{\theta}_{\text{stab}}) \neq \pi^{\text{Min}}(\vec{\theta}_{\text{stab}})$ is equivalent to the existence of multiple intersections of the following functions from $\Theta = [0, 1]$ to $[0, 1]$:

$$f_1(\theta) = \theta^{*-1}(\theta),$$
$$f_2(\theta) = G(w(P(x > \theta | y = 1)) - P(x > \theta | y = 0))$$

Let $Z(\theta) = f_1(\theta) - f_2(\theta)$. Note that $Z(0) = 1 - 0 > 0$ and that $Z(1) = 0 - 0 = 0$. Additionally, consider $Z(\tilde{\theta})$. By the assumptions on $w$, $f_2(\tilde{\theta}) = 1$, and since $f_1(\theta)$ is strictly decreasing in $\theta$ $f_1(\tilde{\theta}) < 1$ so $Z(\tilde{\theta}) < 0$. By the assumptions $Z$ is continuous and thus by IVT $Z$ has at least one additional zero on $[0, 1]$. WLOG let $\theta_{\text{stab}}^{\text{Min}} = 1$ so that $\pi(\theta_{\text{stab}}^{\text{Min}}) = 0$, and let $\theta_{\text{stab}}^{\text{Maj}}$ be the other stable point whose existence we have just proved. Note that $\theta_{\text{stab}}^{\text{Maj}} < \tilde{\theta}$ so by the c-Lipschitz property of $f_1(\theta)$ we have the following:

$$|f_1(\theta_{\text{stab}}^{\text{Maj}}) - f_1(0)| = |f_1(\theta_{\text{stab}}^{\text{Maj}}) - 1| \leq c\theta_{\text{stab}}^{\text{Maj}} \leq c\tilde{\theta}.$$

Since $f_1(\theta_{\text{stab}}^{\text{Maj}}) = \pi(\theta_{\text{stab}}^{\text{Maj}})$ we have $\pi(\theta_{\text{stab}}^{\text{Maj}}) > 1 - c\tilde{\theta} > 1 - c$ which completes part 1 of the proof.

We now complete the second part of the proof. Plugging in $\tilde{\theta}$ to the performative utility and using the assumptions on $G, \epsilon, w$ we have

$$U(\tilde{\theta}) > p_+ * 1 * (1 - \epsilon) - p_- * 0 * \epsilon = p_+(1 - \epsilon)$$

Since any optimal point $\theta_{\text{opt}}$ satisfies $U(\theta_{\text{opt}}) \geq U(\tilde{\theta})$ we know that for all optimal points:

$$p_+ \pi(\theta_{\text{opt}}) P(x > \theta_{\text{opt}} | y = 1) - p_-(1 - \pi(\theta_{\text{opt}})) P(x > \theta_{\text{opt}} | y = 0) > p_+(1 - \epsilon)$$
$$\implies \pi(\theta_{\text{opt}})[P(x > \theta_{\text{opt}} | y = 1) + P(x > \theta_{\text{opt}} | y = 0)] - P(x > \theta_{\text{opt}} | y = 0) > p_+(1 - \epsilon)$$
$$\implies \pi(\theta_{\text{opt}}) > \frac{1 - \epsilon}{P(x > \theta_{\text{opt}} | y = 1) + P(x > \theta_{\text{opt}} | y = 0)} +$$

$$\frac{P(x > \theta_{\text{opt}} | y = 0)}{P(x > \theta_{\text{opt}} | y = 1) + P(x > \theta_{\text{opt}} | y = 0)}$$

Since $0 \leq P(x > \theta_{\text{opt}} | y = 1) \leq 1$ and $0 \leq P(x > \theta_{\text{opt}} | y = 0)$, a simple calculation will show

$$\frac{1 - \epsilon}{P(x > \theta_{\text{opt}} | y = 1) + P(x > \theta_{\text{opt}} | y = 0)} +$$

$$\frac{P(x > \theta_{\text{opt}} | y = 0)}{P(x > \theta_{\text{opt}} | y = 1) + P(x > \theta_{\text{opt}} | y = 0)} > 1 - \epsilon.$$

This in turn implies $\pi(\theta_{\text{opt}}) > 1 - \epsilon$ for any optimal point. Thus all optimal pairs $\vec{\theta}_{\text{opt}} = (\theta_{\text{opt}}^{\text{Maj}}, \theta_{\text{opt}}^{\text{Min}})$ must satisfy $|\pi^{\text{Maj}}(\vec{\theta}_{\text{opt}}) - \pi^{\text{Min}}(\vec{\theta}_{\text{opt}})| < \epsilon$   $\square$

**Remark C.1.** *It is generally possible to construct markets in which both $\epsilon$ and $c$ are small. In fact, the derivative of $\theta^{*-1}(\theta)$ is given by*

$$[\theta^{*-1}(\theta)]' = \frac{\theta^{*-1}(\theta)\phi'(\theta|1) + (1 - \theta^{*-1}(\theta))\phi'(\theta|0)}{\phi(\theta|1) + \phi(\theta|0)}.$$

*Thus, for example any market for which $\phi'(x \mid 1) \approx 0$ for $x < \tilde{\theta}$ and $\phi(x|0) = 1$ can satisfy both. See Figure 4 in appendix B for an example.*

## C.4 PROOF OF THEOREM 3.4

*Proof.* We first prove the statement on stable points. Let $\theta^*(\pi)$ denote the optimal policy for the vanilla utility if the proportion of qualified workers is $\pi$. The existence of the stable set $\vec{\theta}_{\text{stab}}(\theta_{\text{stab}}^{\text{Maj}}, \theta_{\text{stab}}^{\text{Min}})$ such that $\pi^{\text{Maj}}(\vec{\theta}_{\text{stab}}) \neq \pi^{\text{Min}}(\vec{\theta}_{\text{stab}})$ is equivalent to the existence of multiple intersections of the following functions from $\Theta = [0, 1]$ to $[0, 1]$:

$$f_1(\theta) = \theta^{*-1}(\theta),$$
$$f_2(\theta) = G(w(P(x > \theta|y = 1)) - P(x > \theta|y = 0))$$

Let $Z(\theta) = f_1(\theta) - f_2(\theta)$. Note that $Z(1) = 0 - 0 = 0$. Additionally, consider $Z(\tilde{\theta})$. By the strong convexity assumptions, we can directly calculate that:

$$\theta^{*-1}(\tilde{\theta}) = \frac{\phi(\tilde{\theta}|0)}{\phi(\tilde{\theta}|1) + \phi(\tilde{\theta}|0)}$$

Thus, by assumption, $f_1(\tilde{\theta}) > f_2(\tilde{\theta})$ and since these functions are continuous, there must be an additional $\theta'$ such that $f_1(\theta') = f_2(\theta')$, thus the pair $(\theta', 0)$ will be discriminatory.

Next we prove the statement on optimal policies. We will show that if the vanilla utility $U(\theta, \pi)$ is $\gamma-$ strongly concave and that $w$ is small enough, the performative utility will be concave and thus $\theta_{\text{opt}}^{\text{Maj}} = \theta_{\text{opt}}^{\text{Min}}$ since the optimal policy will be unique.

Consider the decoupled performative utility:

$$U_{\text{perf}}(\theta_1, \theta_2) = \pi(\theta_2)\text{TPR}(\theta_1) + (1 - \pi(\theta_2))\text{FPR}(\theta_2)$$

The second derivative of $U_{\text{perf}}(\theta)$ is given by the following:

$$U_{\text{perf}}''(\theta) = \frac{\partial^2}{\partial\theta_1^2}U_{\text{perf}}(\theta_1, \theta_2) + \frac{\partial^2}{\partial\theta_2^2}U_{\text{perf}}(\theta_1, \theta_2)$$

We wish to show that $U_{\text{perf}}''(\theta) < -\gamma'$ form some $\gamma' > 0$. Note that by assumption we have the following:

$$\frac{\partial^2}{\partial\theta_1^2}U_{\text{perf}}(\theta_1, \theta_2) < -\gamma.$$

Letting $\Delta(\theta) = \text{TPR}(\theta) - \text{FPR}(\theta)$

$$\frac{\partial^2}{\partial\theta_2^2}U_{\text{perf}}(\theta_1, \theta_2) = w(wg'(w\Delta(\theta))(\phi(\theta \mid 1) - \phi(\theta \mid 0)) + g(w(\Delta(\theta)))(\phi'(\theta \mid 1) - \phi'(\theta \mid 0)))$$

Because of the boundedness assumptions $g()$ and $\phi()$ we have that

$$|\frac{\partial^2}{\partial\theta_2^2}U_{\text{perf}}(\theta_1, \theta_2)| < 2wK_1K_2$$

Thus if $w < \gamma/2K_1K_2$, the performative firm utility will be strongly concave and thus optimal points will be unique and non-discriminatory. □

**Remark C.2.** *Consider the family of markets with $\phi(x \mid 0) = -ax + \frac{a}{2} + 1$ and $\phi(x \mid 1) = (n+1)x^n$ for some $a$ small and $n$ large. Note that if $p_+ = p_-$ are large enough, this market is strongly concave. Additionally, it is easy to see that there will be a $\tilde{\theta}$ that provides good seperation and additionally that $\phi(\tilde{\theta} \mid 0) \approx 1$ while $\phi(\tilde{\theta} \mid 1) \approx 0$, and thus the lower bound on $w$ in this market can be small as well.*

## D    DISCUSSION ON CONVERGENCE OF RRM (THE MYOPIC FIRMS PROCESS)

For the purpose of this discussion assume that $X \in [0, 1]$. From Perdomo et al. (2020), a myopic firm is sure eventually stabilize if the following conditions are met on the market:

1. $\pi(\theta)$ must be $\epsilon-$ Lipschitz continuous, for some $\epsilon$ not large. Note that:

$$|\pi'(\theta)| \leq w \sup_{c \in \mathcal{C}} g(c) \sup_{x \in [0,1]} |\phi(x \mid 1) - \phi(x \mid 0)|$$

   In general, as long as $g()$ and $\phi(\cdot|y)$ are bounded functions, then this condition will hold for $\epsilon = w * K$. Additionally, if wages are low, $\epsilon$ will be small.

2. The firms non-performative utility $U(\theta)$ and the agents aggregate response $\pi(\theta)$ must be sufficiently smooth. This will follow as long as the functions $G(), \phi(\cdot|y)$ are smooth, which is a technical condition and does not impact the dynamics of the market.

3. The firms vanilla utility $U(\theta) = p_+ \pi \text{TPR}(\theta) - p_-(1 - \pi)\text{FPR}(\theta)$ should be $\gamma-$ strongly concave for all $\pi$. Note that:
$$U''(\theta) = p_+ \pi \phi'(\theta|1) - p_-(1 - \pi)\phi'(\theta|0)$$

   For example, if $\phi'(\theta|1) > \gamma/p_+$ and if $\phi'(\theta|0) < -\gamma/p_-$ then $\gamma-$ strong concavity will be guaranteed.

The most crucial ingredient for convergence is the $\epsilon-$ sensitivity of $\pi(\theta)$. In general, if the function

$$\tau(\theta) = \theta^*(\pi) \circ \pi(\theta)$$

$$\theta^*(\pi) = \arg\max_{\theta \in [0,1]} U(\theta, \pi)$$

has a derivative with upper bound less then one then convergence of RRM will be ensured. This will occur, for example, if the market is regular enough so that $\theta^*(\pi)$ has a bounded derivative and $w$ is small.

## E    EXTENDED RELATED WORKS

**Performative Prediction:** Performative prediction, introduced in Perdomo et al. (2020), seeks to study distribution shifts that are dependent on model deployment. In particular, the reverse causal distribution map considered in this paper leads to a form of subpopulation shift Maity et al. (2021; 2022b). This line of research has been extended to the stochastic optimization setting in the works (Mendler-Dünner et al., 2020; Drusvyatskiy and Xiao, 2020; Wood et al., 2021; Maity et al., 2022a). Stateful performative prediction, introduced in Brown et al. (2022), allows the map $\mathcal{D}$ to depend on both $\theta$ and the current data distribution. The authors of Izzo et al. (2021) and Izzo et al. (2022) propose methods for minimizing the performative risk under a parametric assumption ($\mathcal{D}(\theta) = p(z; f(\theta))$). The work Miller et al. (2021) establishes the necessary conditions for the performative risk to be convex. The authors of Jagadeesan et al. (2022) introduce a zero'th order algorithm for minimizing regret in a performative setting. Our algorithm for anti-causal strategic learning is similar in spirit to the work of Izzo et al. (2021). The main difference is that in our case the performative map is known apriori, which simplifies the procedure and allows for use beyond the parametric setting.

**Strategic Classification:** Although it predates performative prediction, strategic classification Hardt et al. (2015) can be viewed as an instance of performative prediction in which users game their features. The authors of Levanon and Rosenfeld (2021; 2022) give efficient algorithms for learning in general strategic settings. The work Chen et al. (2020) introduces methods for minimizing Stackelberg regret in online strategic classification. The authors of Yu et al. (2022) establish algorithms for a strategic offline reinforcement learning problem. One line of work seeks to study the case where users and the learner do not share all information; the authors of Jagadeesan et al. (2021) assume users view a noisy version of the model; the work Dong et al. (2018) assumes the learner is blind to the strategic agents utility function; the works Ghalme et al. (2021); Bechavod et al. (2022); Barsotti et al. (2022) study the case where the strategic agents must also learn the deployed classifier. The authors of Zrnic et al. (2021) study an extension of strategic classification where the roles are reversed; the agents make strategic decisions before the learner deploys a model. In order to introduce interaction among the strategic agents, the authors of Liu et al. (2022) study a problem where agents compete in contests.

A line of work similar in vein to our work and strategic classification is causal recourse Karimi et al. (2021), König et al. (2023) which seeks to provide actionable intervents for strategic agents to improve their features.

Our work is most aligned with recent efforts to inject causality into strategic classification. Much of the focus has been on improvement (incentivizing agents to improve features in a way that causes responses to improve), beginning with the works Alon et al. (2020); Kleinberg and Raghavan (2020); Haghtalab et al. (2023). The authors of Miller et al. (2020) show that learning good models for improvement is equivalent to solving causal inference problems. A follow-up line of work considers this problem in specific scenarios; with Shavit et al. (2020) focused on regression, and Harris et al. (2022) on instrumental variables. The work Mendler-Dünner et al. (2022) reframes performative predictions as causal interventions. A closely related work is Horowitz and Rosenfeld (2023), which gives a method for minimizing the causal strategic empirical risk.

**Economic models of labor markets:** Studies of statistical theories of discrimination began with the two lines of work (Arrow, 1971; Phelps, 1972). In Phelps (1972), discrimination in labor markets is due to two groups of workers being exogenous, while Arrow (1971) shows that even if groups are endogenous, discrimination can occur in equilibrium. These ideas were developed by Coate and Loury into a labor market model in (Coate and Loury, 1993). More recently, the authors of Moro and Norman (2004) and Moro and Norman (2003) extended this model by including interactions among groups of workers and a wage structure set by inter-employer competition. The line of work Fryar et al. (2008); Fryar and Loury (2005; 2013); Craig and Fryer (2017) studies the impacts of different afemployerative action policies (*i.e.* color blinded vs color sighted) in a variety of market settings including those similar in spirit to (Coate and Loury, 1993). The more contemporary works Liu et al. (2020); Somerstep et al. (2023) utilize the Coate and Loury model to study discrimination in algorithmic decision making. For a survey on statistical discrimination in labor market, see (Fang and Moro, 2011a).

