# OpenReview forum: "Learning in reverse causal strategic environments with ramifications on two sided markets"
_ICLR.cc/2024/Conference — ICLR 2024 poster_

### Official Review · Reviewer_L1Mv · 2023-10-31

**Soundness:** 2 fair
**Presentation:** 2 fair
**Contribution:** 2 fair
**Rating:** 6
**Confidence:** 3

**Summary:**

* The paper introduces a reverse-causal strategic classification setting, and analyzes it primarily within the context of the Coate-Loury labor market model.
* In the Coate-Loury labor market model, an employer trains a screening function $f(x)$ to detect high-skill workers ($y=1$), and workers respond strategically by possibly increasing their skill level at cost $c$. Worker features $x\\in[0,1]$ are a noisy function of skill ($x\\sim\\Phi(\\cdot | y)$), making the strategic response reverse-causal.
* The analysis investigates the gap between the associated repeated risk minimization (RRM/”stable”) and performatively optimal (PO/”performative”) policies (eq. (2.2) and eq. (2.3), respectively).
* For a Coate-Loury model with a uniform decision rule (same decision threshold $\\theta$ for all population), Theorem 3.1 gives conditions under which a PO policy results in a higher proportion of skilled workers and increased employer’s utility compared to RRM.
* For a model with non-uniform decision policies (two groups {Maj,Min} and a separate decision threshold for each), Theorem 3.2 gives conditions under which RRM leads to a discriminatory population composition, and PO leads to population composition which is approximately balanced.
* Finally, a generalization of the Coate-Loury model with inter-group interactions is investigated by numerical simulation. Results indicate that a PO policy leads to higher employer utility, but can generally reduce the welfare of the workers.

**Strengths:**

* Problem is well-motivated. Theoretical framework is interesting.
* Performative response is derived directly from an established economic model.
* Running examples in the introduction aid understanding, and strengthen applicability.
* Model assumptions are presented clearly, and relaxed gradually.

**Weaknesses:**

* Soundness concern: Repeated risk minimization (RRM) plays a central role in the theoretical analysis (eq. (2.2) ,eq. (3.1)), and convergence is claimed to be due to “repeated risk minimization, which is known to converge to performatively stable policies (Perdomo et al., 2020)”. However, if I understand correctly, the convergence guarantees in Perdomo et al. 2020 rely on strong regularity assumptions (e.g., $\\beta$-joint smoothness, $\\gamma$-strong convexity, $\\varepsilon$-sensitivity) and RRM can fail to converge in their absence (see Theorem 3.5 and Proposition 3.6 in Perdomo et al. 2020). I was unable to find a discussion of these assumptions and their applicability in the paper, and therefore it is not clear why RRM is guaranteed to converge in this context.
* The learning setting is unclear (see questions below).
* Code is not provided, making it hard to validate and reproduce the results of Section 4, which rely on numerical evaluation.
* Interaction model assumes one dimensional features and strict monotone likelihood ratio. It is not clear how results extend to higher-dimensional features and more complex distribution structures.

**Questions:**

* RRM convergence: How does the claim about convergence to performative stability relate to the formal guarantees given by Perdomo et al.?
* Additional related results in Perdomo et al.: In the paragraph below the statement of Theorem 3.1, it is claimed that “Theorem 3.1 gives conditions for there to be an appreciable gap. This complements prior results (for example, in Perdomo et al. (2020)) that provide conditions under which the gap is small.”. In contrast, Theorem 4.3 in Perdomo et al. 2020 predicts that the gap between the PO and RRM policies is expected to be small. What is the relation between the gaps presented in this paper and Theorem 4.3 in Perdomo et al.? If some required Theorem 4.3 are not met, which ones? And how does it relate to the RRM convergence guarantees discussed in the question above?
* Learning setting: At what stage data is available to the employer, and how do they learn from it? How do the main results extend to scenarios where predictors are learned from finite datasets?
* Do similar results hold for the content creation scenario described in Example 2.2? What would be required in order to apply the results in other scenarios?
* Small question about notations: What is the difference between $w\\int_X 1_{\\{f(x)=1\\}} d\\Phi(x|Y=1)$ and $\\int_{[0,1]} w f(x) d \\Phi(x|1)$ in Example 2.1?
* Small typo in Section 2: examplse. Appendix has undefined references: (??).

---

> ### Author Response · Authors · 2023-11-15
> **Rebuttal for Reviewer L1Mv**
>
> We thank the reviewer for the helpful comments, we address particular concerns below.
>
> *Soundness concern: Repeated risk minimization (RRM) plays a central role in the theoretical analysis (eq. (2.2) ,eq. (3.1)), and convergence is claimed to be due to “repeated risk minimization, which is known to converge to performatively stable policies (Perdomo et al., 2020)” \ldots  I was unable to find a discussion of these assumptions and their applicability in the paper, and therefore it is not clear why RRM is guaranteed to converge in this context.*
>
> In response to this soundness concern we have made the following changes to the submission:
>
> 1. We have re-written the discussion of performative prediction in the Coate-Loury model  (section 3.1) in order to clarify that our results DO NOT depend on the convergence of RRM. Each theorem is a comparison of the impacts of the two types of solutions in performative prediction (stable and optimal points) on labor markets, and for such a comparison to be valid we only need that both solution types exist, which is always guaranteed. We also point out that  the conditions provided in (Perdomo et al. (2020)) are sufficient, but not necessary, and our empirical results demonstrate that in practice the convergence of RRM/reactive firms is not an issue in a wide range of markets.
>
> 2. A more thorough discussion on the market conditions needed to guarantee RRM convergence is added to the appendix (see appendix D). The primary benefit of imposing such conditions on the market is that all ``reactive" employers are guaranteed to eventually stabilize. This is not needed for our results but is still nice conceptually. Given this, we have added two new theorems (now theorems 3.2 and 3.4) which provide results in markets which are compatible with these conditions (particularly we show our main take aways are similar in low wage markets).
>
> *Additional related results in Perdomo et al.: In the paragraph below the statement of Theorem 3.1, it is claimed that “Theorem 3.1 gives conditions for there to be an appreciable gap. This complements prior results (for example, in Perdomo et al. (2020)) that provide conditions under which the gap is small.”. In contrast, Theorem 4.3 in Perdomo et al. 2020 predicts that the gap between the PO and RRM policies is expected to be small. What is the relation between the gaps presented in this paper and Theorem 4.3 in Perdomo et al.? If some required Theorem 4.3 are not met, which ones? And how does it relate to the RRM convergence guarantees discussed in the question above?*
>
> None of the conditions in (Perdomo et al. (2020)) are broken, but the ``$\epsilon$-sensitivity condition" is only satisfied for a large value of $\epsilon$ when the $w$ is large. Thus the upper bound on the gap between PO and PS policies from (Perdomo et al.(2020)) is also large under the conditions of our theorems 3.1 and 3.3. Appendix D now clarifies the importance that wage plays in convergence of RRM.
>
> *Learning setting: At what stage data is available to the employer, and how do they learn from it? How do the main results extend to scenarios where predictors are learned from finite datasets?*
>
> In a stochastic (or finite data) setting the order of learning at time $t$ is as follows:
>
> 1. The firm deploys decision $\theta_t$
>
> 2. Firm recieves data (and corresponding loss/reward) drawn from $\mathcal{D}(\theta_{t})$. The firm may use this to either deploy a reactive decision or in some algorithm that converges towards an optimal policy (the discussion on this in appendix A assumes a finite data set)
>
> Although we work in the population setting, we expect similar results in the finite-sample setting in which the firm has to learn its predictor from data. Under standard assumptions that guarantee generalization, the firm's learned predictor will be close to its optimal predictor in the population setting that we study.
>
> *Do similar results hold for the content creation scenario described in Example 2.2? What would be required in order to apply the results in other scenarios?*
>
> This is an intersting line of follow up research, the content creation game is quite different, as the ``content creators" actually compete with one other for user attention. This means the response of the creators are participants in a game and their response will be some type of equilibria of the game. This type of agent response is very different from the agent best response that we study.
>
> *Small question about notations: What is the difference between $w\int_X 1_{f(x)=1}d\Phi(x|1)$ and $w\int_{[0,1]} 1_{f(x)=1}d\Phi(x|1)$ in Example 2.1?*
>
> This is a typo, $X$ should be [0,1] in this case.
>
> *Code is not provided, making it hard to validate and reproduce the results of Section 4, which rely on numerical evaluation.*
>
> We have now uploaded our code.

---

> ### Author Response · Authors · 2023-11-20
> **Follow up with reviewer L1Mv**
>
> Dear reviewer L1Mv,
>
> Thank you for taking time to review our paper and provide valuable feedback. Hopefully you have had time to read our response. Please let us know if you have further questions or comments, we would be happy to address them. If you found our responses and updated submission satisfactory, please consider updating your score.

---

> > ### Comment · Reviewer_L1Mv · 2023-11-21
> >
> > Thank you for the detailed response, and for the revision. The added discussion regarding RRM convergence, the clarifications regarding the gap between PO and PS policies, and the added code are all very helpful. I increase my overall rating to 6.
> >
> > I believe the paper would benefit from a thorough discussion of limitations, particularly with respect to the applicability of assumptions, and the case of finite data sets: Regarding applicability, I wonder if the Coate-Loury literature contains works in which the model is fitted to actual data, providing evidence to support the assumptions made about parameter relationships, regularity, etc. For the finite data case, I believe the paper would also benefit from a more detailed discussion of the finite data setup, limitations of the proposed methods, and remaining questions.
> >
> > Also note that the revised manuscript appears to contain minor typos - in the paragraph after the statement of Theorem 3.1 ("Theorems 3.1 and 3.1" - appears twice), and in the last equation of Example 2.1 (uppercase Y in the first integral, lowercase y in the second integral).

---

> > > ### Author Response · Authors · 2023-11-29
> > > **Follow up with Reviewer L1Mv**
> > >
> > > We thank the reviewer for taking the time read our second submission, revise their score and provide us with more helpful feedback. In an effort to incorporate this feedback, we have uploaded a new draft with the following additions:
> > > 1) Appendix A now includes a more thorough discussion on the learning setting, and the assumptions that is required for the algorithm.
> > > 2) We have added a small paragraph to the end of section 3.1 which discusses ``applicability" of our market assumptions.

---

### Official Review · Reviewer_jt3C · 2023-11-06

**Soundness:** 3 good
**Presentation:** 3 good
**Contribution:** 3 good
**Rating:** 8
**Confidence:** 4

**Summary:**

This paper proposes a novel formulation of the causal strategic classification problem. Here, there is a set of agents, each described by x, the values for a set of features and with a true label y. A classifier wishes to classify agents based on their features x to minimize some loss function defined w.r.t. the true labels y of the agents.

However, the agents are strategic, and given a classifier with parameters \theta, can change their label y by changing their features at a certain cost, in order to obtain a utility from the classification. The problem is motivated through the running example of the labor market, where the employer has a hiring policy based on the skill of the worker. Workers can change their skill in order to increase their chances of being hired and obtain the wage from being hired.

The model assumes that there is a (reverse) causal model that determines the relationship between the features and the label (or vice-versa respectively). The paper focuses on the setting with a reverse causal model. Here, agents are allowed only to determine whether they want a change in their label, and the reverse causal model determines how the features must be changed.

The technical results of the paper deal with analyzing the solutions of the game that results from the interactions between the classifier and agents under such a reverse causal model, with interesting results and consequences for a well-studied model of the labor market. The paper provides interesting results on the effect of a reactive classifier (which iteratively updates the parameters \theta after the agents respond to the classifier used in the previous iteration) and a strategic classifier which anticipates the best response of the agents on the welfare of the classifier and agents.

**Strengths:**

- First, I found the model very interesting, and the main technical contribution of the reverse causal setting for performative prediction to be interesting and significant. The topic is clearly relevant to ICLR, and similar models may be relevant several other applied fields.
- The main technical results appear sound, and I appreciate the nice discussions following the theorems discussing how to interpret the results and the implications for the running example of the labor market.
- The findings that the presence of a performative classifier can hurt agent welfare and that its fairness properties are brittle are also interesting.

**Weaknesses:**

- No major weakness. The authors may consider doing a more thorough pass to fix some minor typos.

**Questions:**

None

---

> ### Author Response · Authors · 2023-11-15
> **Rebuttal for Reviewer jt3C**
>
> We thank the reviewer for the encouraging words on our work, and alert them that in response to another reviewer a second version of the submission has been posted.

---

### Meta-Review · Area_Chair_zXCA · 2023-12-12

**Metareview:**

This paper looks at strategic classification, where agents controlling inputs can change subsets of those inputs' features so as to adversarially misclassify themselves, but at some cost.  Reviewers appreciated the connection to known theoretical labor market models, but were worried about the lack of experimental validation in models more closely mimicking reality, convergence issues, and more generally relations to the larger literature.

**Justification For Why Not Higher Score:**

Strong reviewer scores, but missing at least one review.  Also, very strong assumptions of causality -- assumptions go a long way with papers like this, and this one is not particularly realistic.

It's not clear to me why this paper belongs at ICLR vs. an EC style of conference.  This feels very niche and involves very little machine learning.

**Justification For Why Not Lower Score:**

Strong reviewer scores, including after the rebuttal.  There is a history of market design papers within the ML community, but relatively less so at ICLR vs NeurIPS/ICML.

---

### Decision · Program_Chairs · 2024-01-16

Accept (poster)